# Developing STEM Career Identities among Latinx Youths: Collaborative Design, Evaluations, and Adaptations during COVID-19

**DOI:** 10.3390/bs13110949

**Published:** 2023-11-18

**Authors:** Chong Myung Park, Hayoung Kim Donnelly, Angelica Rodriguez, Luis Esquivel, Cecilia Nardi, Paul Trunfio, Alexandra Oliver-Davila, Kimberly A. S. Howard, V. Scott H. Solberg

**Affiliations:** 1Wheelock College of Education and Human Development, Boston University, Boston, MA 02215, USA; lesquive@bu.edu (L.E.); khoward@bu.edu (K.A.S.H.); ssolberg@bu.edu (V.S.H.S.); 2Department of Psychiatry, University of Pennsylvania, Philadelphia, PA 19104, USA; hayoung.donnelly@pennmedicine.upenn.edu; 3Sociedad Latina, Boston, MA 02120, USA; arodriguez@sociedadlatina.org (A.R.); alex@sociedadlatina.org (A.O.-D.); 4Office of Government & Community Affairs, Boston University, Boston, MA 02215, USA; cnardi@bu.edu; 5Department of Physics, Boston University, Boston, MA 02215, USA; trunfio@bu.edu

**Keywords:** youth, STEM career identity, COVID-19, Latinx, community-based participatory action research, network science

## Abstract

In response to the low representation of Latinx adults in STEM occupations, this community-based participatory action research study aims to increase the number of middle school youths developing STEM career identities and entering high school with the intention to pursue STEM careers. The students were provided with summer and after-school activities focusing on network science and career development curricula. Using a quasi-experimental pretest–posttest design and career narratives, this study examined the changes in STEM and career self-efficacy, as well as career identity. The results show improvements in self-efficacy, an increased number of youths with intentions of pursuing future STEM career opportunities, and deeper reflections on their talents and skills after program participation. This paper also describes the program development and implementation in detail, as well as the adaptations that resulted from COVID-19, for scholars and educators designing similar programs. This study provides promising evidence for the quality of STEM and career development lessons in supporting the emergence of a STEM career identity and self-efficacy.

## 1. Introduction

### 1.1. Developing STEM Career Identities among Latinx Youths: Impact of a Pilot Intervention

The United States is in the midst of the fourth Industrial Revolution [1]. With technological advances prioritizing a range of STEM-related careers, increasing the number of youths from under-represented racial and ethnic groups, as well as females, who want to pursue STEM careers continues to be a critical social justice challenge. While representing 16% of the overall workforce, Latinx adults are severely under-represented in the science and technology fields, with only 7% participating in a STEM occupation [2]. The Latinx population includes diverse cultures and countries of origin and, as a group, it is the largest ethnic or racial minority in the United States population, estimated at over 57 million in 2016 [3]. From 2000 to 2018, the average college enrollment rates for Latinx youths in the U.S. increased from 22% to 36% [4]. Still, the college enrollment rates among Latinx youths remain lower than those of White (42%) and Asian (59%) youths in the U.S. [4]. In 2019, the percentage of Latinx people who had attained either a two- or four-year post-secondary degree was low at 31% when compared to Asians (78%), Whites (56%), and Blacks (40%) [4]. Since 2015, young Latinx adults have continued to be under-represented among STEM bachelor’s degree recipients [5].

This study reports on the design and evaluation results of a summer and after-school STEM enrichment program for middle school-aged (aged 12–15) Latinx youths that aims to increase aspirations to pursue STEM careers by combining advanced data science STEM learning with career development activities. This study focuses on three elements of the program. The first element involves strategic efforts to establish an effective community-based participatory action research collaboration [6,7]. The second element involves the design of STEM and career development lessons, which involves civic engagement and role model sessions within the context of remote learning due to the COVID-19 pandemic and the transition back to in-person learning. The COVID pandemic left a profound impact on schools and the workplace due to increased digitalization across these settings [8]. The third element involves assessing the progress made during the first two and a half years of implementation.

The three elements mentioned above are known to increase school success and post-secondary engagement among Latinx youths by increasing social capital through connecting with role models and mentors and helping youths establish and develop positive cultural identities and post-secondary self-efficacy [9,10,11,12]. These elements are also associated with building the resiliency and coping skills needed to effectively manage structural inequities and challenges that would otherwise lead to risky behaviors and academic disengagement [11].

Rooted in a self-concept that focuses on perceptions of one’s current abilities that are more stable [13], this study examines self-efficacy that focuses on one’s perceptions of competence to perform tasks in the future in specific situations/contexts. Therefore, self-efficacy in this study is defined as one’s perception of success in specific domains, particularly STEM. As self-efficacy shapes career interests, goals, and eventual career identity [14,15], those with higher career self-efficacy are likely to exhibit greater identity achievement [16]. Career identity in this study is defined as the possession of a clear picture of one’s goals, talents, and skills [17,18].

### 1.2. Theoretical Framework: Community-Based Participatory Action Research

The program was implemented as a community-based participatory action research (CbPAR) effort between Sociedad Latina, a youth-serving organization, and Boston University, a higher education institution. Using Wilson’s [19] framework, CbPAR in this study (a) aims to promote social transformation for the target population, (b) engages a team of stakeholders in decision making from the initial phase, (c) identifies a shared perspective of the issues that need to be addressed, and (d) designs and implements innovative practices that not only meet the needs of the target population but also empower them and increase the capacity of the organization [19,20]. Sociedad Latina is considered a successful and well-respected youth-oriented organization that serves students and their families in the Boston area. Sociedad Latina is recognized for combining enriched academic learning opportunities with cultural awareness and pride, creative arts, civic engagement, and workforce readiness activities and services [21]. Starting with middle school Latinx youths, Sociedad Latina provides enrichment and mentoring support throughout high school and into their first two years of college. They report that all of their students are graduating from high school and successfully entering post-secondary education. Given the tremendous impact Sociedad Latina is having on youths and the community, the project team worked to build capacity among their STEAM (science, technology, engineering, art, and math) team to offer advanced STEM and STEM career development activities. The project goal was to increase the number of Latinx middle school youths who enter high school with expressed intentions to pursue a STEM learning and career pathway.

According to the National Research Council [22], out-of-school and after-school STEM learning opportunities are particularly valuable when youths receive access to high-quality and challenging STEM learning experiences that otherwise may not be available, especially within under-resourced schools in low-income communities. After-school STEM learning activities are most effective when they offer engaging lessons, respond to the cultural and linguistic contexts of youths, and help youths become aware of the relevance of different STEM learning opportunities to themselves and their futures [22]. Expanding access to quality STEM learning is especially important for Latinx youths because they are less likely to have access to educators who received a post-secondary STEM degree [23]. When STEM learning is offered by community-based, youth-serving organizations such as Sociedad Latina, youths will be more likely to receive culturally and linguistically responsive mentors and learning experiences. A qualitative assessment of a summer STEM enrichment program found that more youths considered STEM career options after the program, and most importantly, there was also evidence that students of color developed a stronger cultural identity [24].

### 1.3. Strategies for Establishing Collaborative Research

As a community-based participatory action research effort, principal investigators and staff from Sociedad Latina and Boston University sought to weave various areas of expertise into a coordinated plan of action to design, deliver, and evaluate the STEAM (science, technology, engineering, art, and math) and career development lessons. There are three elements that distinguish participatory action research from conventional research: (1) it aims to foster and enable action that involves an iterative process of reflection and action; (2) it deliberately shares power among participants, including both the researchers and the researched; and (3) it does not consider the participants as “subjects’’ to investigate but promotes the active participation of the researched [6]. As such, the collaboration was iterative in nature and facilitated youth participation in commenting on the design and content of lessons and helped them create their own projects to reflect on what they learned from the program.

Even given the best intentions, there are differences in power and privilege that need to be considered and managed [25]. To address these issues, a multicultural perspective was used, whereby the two organizations, as well as the multiple units within Boston University, were each treated with a different worldview. For example, it was assumed that each organization and unit within Boston University entered the collaboration with different, albeit complementary, motives for engaging in the project. Sociedad Latina, for example, has already established a history of success in serving immigrant Latinx youths and their families. Working with a university was perceived as having the potential to elevate the quality of the work already taking place, as well as engaging in evaluation strategies that would further communicate the impact of their efforts. As such, the collaboration could potentially support the visibility and reputation of Sociedad Latina, which could translate into funding to sustain and expand existing programs. The motivation of the faculty from Boston University’s Department of Physics and College of Education was to (1) test whether the combination of advanced STEM Learning and career lessons could result in the establishment of a STEM career identity and describe the conditions that were associated with this emergence, and (2) collaborate with a community organization to “incubate” innovative STEM and career activities, with the goal of sustaining collaboration with Sociedad Latina to expand such partnerships with other organizations and schools to implement the activities. Boston University’s Government and Community Affairs (GCA) staff were motivated by the need to leverage ways for the university to support our local Boston community. In lieu of property taxes, large private universities are under pressure to demonstrate a “return on investment.”

To further manage differences in power and privilege, deliberate and strategic efforts were made to understand how our respective identities and relationships with the community influence the research and collaboration and to help each of us reflect on our experiences [25]. For example, two common characteristics of university-led community projects include subcontracting with local organizations and conducting activities on campus. Subcontracting to youth-serving organizations holds power within the university, which could decide to use the funds in other ways or select a different organization to work with altogether. Fortunately, the National Science Foundation, which funded this project, allowed both organizations to submit separate budgets as part of a “collaborative” proposal. As a collaborative proposal, Sociedad Latina controlled its own budget. Rather than creating a new on-campus after-school program, Boston University’s project leaders relied on the expertise of Sociedad Latina staff to learn how to translate STEM and career activities for middle school youths. While Boston University provided evidence-based content ideas for the lessons, Sociedad Latina translated the lessons in a manner that was culturally and linguistically responsive to middle school Latinx youths.

In reflecting on the nature and value of the collaboration, Sociedad Latina staff reported that coordination and shared ownership were established through weekly check-ins, collaboration on curriculum design, and support for content focused on Network Science and Career Development. Furthermore, staff reported that the collaboration has helped them learn about interactive online STEM and career development resources for youths. It was felt that the collaboration supported both the STEAM team and Sociedad Latina as a whole. For example, “Boston University has participated in organization-wide events, such as our remote VIVA Concerts and our Summer Celebration, and served as a strong advocate in supporting our youth, one of the most evident values of our collaboration.”

The funding announcement in the spring of 2020 coincided with the first wave of COVID-19 in the United States. The impact of COVID-19 included a later start to the summer program, a reduction in the program’s time and participation, and the need to adapt what was intended for in-person learning to remote learning, as well as a shift to online surveys for evaluation. As a result, the initial summer program served as a pilot program in order to design and test innovative, interactive STEM and career development activities that could be offered in a virtual small group environment. The program went through several iterations over the next two years.

## 2. Method

### 2.1. Participants

A total of 190 Latinx middle school students participated in the 2020 summer, 2020–2021 academic year, and 2022 summer programs and the two types of evaluations: one that examined lesson quality and another that focused on youths’ career identities. Participation rates were significantly reduced due to the remote learning nature of the program and the fact that many families did not have the technology needed to access the program. All of the youths were also English Language Learners (ELL) and attended urban middle schools with high concentrations of Latinx ELL youths. The surveys were collected anonymously using pre-assigned, randomized ID numbers; therefore, the data cannot be identified by gender or age.

### 2.2. Designing the STEM and Career Development Lessons

Referred to as “STEAM” (adding Arts to STEM), this collaboration represents one of many out-of-school, summer, and after-school programs offered by Sociedad Latina. Boston University offered three strands of work and resources important to this collaboration. First, advanced network science lessons were made available by the Physics Department faculty through a national “NetSci for All” project that offers a number of interactive and online activities for pre-college youths [26]. Second, middle and high school career development lessons, which are collectively referred to as My Career and Academic Plan (MyCAP) in Massachusetts, and program evaluation strategies were made available by the College of Education faculty. And third, Boston University’s Government and Community Affairs (GCA) staff connected STEM role models with the program and leveraged additional university resources such as IT consultation support when the shift to remote learning occurred due to COVID-19.

Drawing from Social Cognitive Career Theory (SCCT) [14,15,27], this program focused on designing network science and career development lessons that were expected to (a) increase STEM learning and STEM career exploration self-efficacy, and (b) facilitate the emergence of a STEM career identity (i.e., youths stating their intention to pursue future STEM learning and career opportunities). The SCCT model has been further tested, indicating that career exploration fosters career decidedness self-efficacy [15] and self-efficacy and outcome expectations predict goals, which, in turn, predict actions [27]. Life satisfaction was also predicted by self-efficacy and support [28]. The network science career development lessons were designed in accordance with research identifying key ingredients for effective lesson design [29,30], as well as culturally responsive career development activities [31,32]. Up to five key ingredients were incorporated into each lesson, including (1) engaging in mastery learning opportunities, (2) caring and encouraging adults, (3) making connections between the skills being learned and the world of work, (4) engaging in personalized assessments that provide individualized results, and (5) offering an immersive world of work experience with professional role models. Each lesson was presented in English and Spanish and was facilitated by Latinx role models who expressed strong values and encouragement regarding STEM learning and careers. Previous research indicates that engaging in STEM career interventions using these ingredients is an effective way to increase middle school Latinx youths’ STEM career decision-making self-efficacy and STEM self-efficacy [33].

#### 2.2.1. Network Science

Our focus on middle school Latinx youths is responsive to longitudinal research indicating that in order to pursue STEM careers, youths should be leaving middle school with the math and science skills needed to pass calculus and physics prior to high school graduation [34]. In addition, longitudinal research indicates that youths from other under-represented minority groups rarely consider STEM careers unless learning opportunities are specifically designed to help encourage STEM learning [35,36]. We focused on creating access to the field of network science for three reasons: (1) the core skills of network science—data analysis, visualization, and network maps—help students see problems from the point of view of how the elements of the problem are connected at a systems level, (2) there is a high demand for data scientists in the Boston area as well as in many areas of the United States, and (3) the broad range of data science skills transfers to multiple pathways to high-paying STEM and non-STEM occupations [37]. Glass Technologies reported that in 2015, data science skills were aligned with over 2 million job openings and represented over 50 occupations across a wide range of sectors, including finance, insurance, IT, manufacturing, and professional services such as human resources and marketing.

While network science is not a typical science subject taught in middle and high schools (c.f., biology, chemistry, physics), it encompasses all science subjects through the concept of network thinking. The United States National Research Council provided a definition of network science: “the study of network representations of physical, biological, and social phenomena leading to predictive models of these phenomena.” For instance, traffic is a network of transportation heading to different directions/destinations, and music is a network of musical instruments making different sounds, beats, and rhythms. The network science curriculum covers several science subjects, including biology, chemistry, physics, and computer and data sciences. While it does not focus on a particular science domain due to its broad connectivity with other science subjects, the network science curriculum has implications for all science subjects. Network science can also be a good introduction to STEM for younger students to show how science concepts are connected to our everyday lives, thereby fostering interest in STEM.

A total of 5 network science problem-based lessons [26] were offered during the first summer pilot phase, which expanded to 15 lessons over the next two years (see Appendix A for a list of the network science lessons). Using the basic STEM principles, such as water in chemistry, circuits in physics, and viruses in biology, the lessons were extended to “network thinking” and its applications. Below are the five pilot lessons that became foundational for creating additional lessons. Lesson 1 focused on building network thinking through guided simulations of the flickering hydrogen bond network in liquid water, with the aim of helping students appreciate that networks are not just familiar social ones but are at the core of the most common liquid on Earth, essential for life. Through this lesson on interacting water molecules, students learned how to define the nature of a network, describe what types of networks exist in everyday life, and understand how network maps are connected to data. Lesson 2 focused on the exploration of elementary electric circuit theory as a metaphor for network thinking and simultaneously learned about real electric circuits. The intent was to help students build their own electrical networks and analyze the flow of electrons as information and energy carriers. Students learned how electrons freely move in conductors and how this random motion of electrons (electrical charge) becomes an electrical current once placed in a circuit consisting of light bulbs, wires, batteries, and resistors. Using a model of electric circuits and a basic understanding of Ohm’s Law, students were able to translate this knowledge into networks and their applications to other systems, such as the parallel and series resistor network models in human physiology or models of traffic flow. Lesson 3 focused on disease and epidemiological models and their application to COVID-19. This real-world problem facilitated students’ engagement in different network scenarios. An online epidemic game titled VAX! was used to help students understand how the virus spreads and how to prevent the virus from spreading [38]. Lesson 4 focused on students building their own datasets through an activity, where the data were obtained from their personal relationships (e.g., who they talk to and are connected with) by entering the information into a spreadsheet and then creating a graphic visualization. “Gephi” visualizer (Gephi 0.10 [Computer software]) [39], and Kumu [40] provided the results that showed how the students were connected to their larger network. In Lesson 5, students presented what they had learned about network science and had an opportunity to meet with a professional biomedical researcher from Boston University’s National Emerging Infectious Diseases Lab (NEIDL). The virtual presentation modeled the safety standards used to put on a laboratory suit and conduct research tests on disease samples such as COVID-19.

#### 2.2.2. MyCAP Career Development Lessons

To facilitate the formation of a STEM career identity, youths participated in a series of My Career and Academic Plan (MyCAP) [41] lessons that explicitly explored how the NetSci skills they were learning could be considered an emerging talent that is transferable to a wide range of STEM and non-STEM career options [20]. MyCAP is the Massachusetts term for an Individualized Learning Plan (ILP), which is a national college and career readiness effort that engages youths in personalized career and academic planning. While ILP is a national term, the majority of states like Massachusetts have selected alternative naming conventions, such as Academic and Career Plans (Wisconsin) or Individual Career and Academic Plan (Colorado, Oklahoma). ILPs/MyCAP offer a positive youth development paradigm shift for Latinx youths that de-emphasizes “career choice and decision-making” and focuses on student-driven and personalized learning, with an emphasis on their talents and future goals [42,43]. As a process, ILPs vary from the traditional career education paradigm by placing an emphasis on positive youth development. Sunny Hansen [44] described this as “career development education,” which was defined as

“A developmental process by which individuals have an opportunity, through a systematic and sequential set of experiences, to know themselves better, to know their environment (options) better, and to act on that knowledge more purposefully and creatively (p. 42).”

ILP activities offer youths opportunities to become aware of their emerging talent (self-exploration skills), identify occupational opportunities that align with their emerging talents (career exploration skills), develop academic course plans, and consider early college and certifications, as well as post-secondary pathways that align with their future careers and life goals (Career planning and management skills) [20]. 

With respect to evidence supporting the impact of engaging in ILPs on positive youth development, a multi-method, multi-study effort found a number of promising results using case studies, correlational methods, and focus groups [45]. For example, focus groups with youths, families, and educators from 14 high schools in four states found that engaging in ILPs was believed to have increased student engagement, including higher grades and rigor in course selection [45]. District and state education leaders reported that traditional teaching silos between general education and special education were breaking down as ILPs helped general education teachers recognize the potential of students with IEPs to pursue post-secondary and career goals. The strongest evidence for the value of ILPs was a quasi-experimental study conducted by the Singapore Ministry of Education. Using a nationally stratified randomized sample, the ministry found that exposure to four ILP lessons was associated with increased career search self-efficacy as well as more engagement in career exploration [46].

The MyCAP portion of the summer pilot program involved three 45-minute lessons that were adapted from Making My Future Work (MMFW), which offers a rich resource of evidence-based career development activities [47,48]. Lesson 1 was adapted from MMFW’s Who Am I lesson to support students’ self-exploration of their emerging talents. The lesson was further adapted to emphasize the discovery of their deep human skills (i.e., advanced social-emotional learning skills) [49], which employers believe are critical workforce readiness skills needed to compete in the fourth Industrial Revolution [50]. Using the Collaborative for Academic, Social, and Emotional Learning (CASEL) framework aligned with the employer-identified workforce readiness skills [51], the students selected from 31 deep human skills that were organized around five themes: Positive Impact, Communication, Listening Skills, Dependability, and Organizational Skills. The students were asked to indicate the skills they believed they most clearly possessed, with the goal of using this information to create their unique, personal brand. In the second lesson, students were introduced to the O*NET website, which provides information on skills and resources in different occupations. The students took the skills and adjectives from the first lesson and looked at the advanced skills and careers aligned with their existing skills and adjectives. In week 3, students participated in an online scavenger hunt that asked them to independently navigate the STEM career cluster found within the O*NET website, compare and contrast different STEM occupations, and learn about their average salaries and required education levels. The students reported back on what they had learned. Through Lesson 3, students became aware of the transferability of STEM skills to the world of work. Students were encouraged to identify the network science skills that they had learned such as complex problem solving, systems analysis, programming, critical thinking, mathematics, etc. This assessment then provided an interactive list of hundreds of occupations that aligned with their emerging network science skills. Based on these three foundational lessons, the career lessons continued to expand over the next two and a half years by adding elements focusing on optimizing youth learning, personal roadmaps, modified RIASEC assessments that focus on self-awareness, and support networks (see Appendix A for a list of career lessons).

#### 2.2.3. Adapting the Curriculum to Grade Levels and COVID-19

The science curriculum was originally designed for high school students but was adapted and redesigned for middle school youths with an emphasis on ensuring that the materials were age-appropriate and translated into Spanish. There was also a need to translate in-person classroom lessons into interactive online lessons that could be delivered in remote learning environments. To promote active engagement and provide targeted lessons, Sociedad Latina staff created interactive presentations that could be implemented effectively using web-based technologies (e.g., Zoom, WebEx) and subsequently incorporated other interactive technology strategies such as Nearpod [52]. Their design efforts allowed youths to personalize their experiences and expand their learning by sharing with a mentor and a small group of peers. The challenging learning environment as a result of the pandemic called for frequent check-ins with the youths and their families regarding their immediate needs to continue their educational experiences. One of the areas the youths and their families were struggling with as they transitioned to online learning was technology, such as accessing WiFi and troubleshooting Chromebooks, which Boston Public Schools provided to students. While the school district helped ensure that families had access to the Internet and a laptop computer, there was often not enough guidance and information for the youths and their families to navigate the remote learning environment. To support Sociedad Latina’s efforts in supporting the IT needs of these families, Boston University’s Government and Community Affairs staff successfully leveraged the institution’s Information Systems and Technology (IS and T) staff to provide consultation support.

This study aimed to increase the number of middle school youths developing STEM career identities and entering high school with self-efficacy in STEM and potentially with intentions to pursue STEM careers. The specific research questions included the following: (a) To what extent did the middle school youths develop self-efficacy in STEM through the network science curriculum? (b) To what extent did the middle school youths develop self-efficacy in STEM career development? (c) How did the middle school youths’ career identities change before and after the program?

### 2.3. Data Collection Procedure

The project objectives consisted of assessing whether the STEM and MyCAP lessons showed promise in supporting the emergence of a STEM career identity and self-efficacy. Youths participated in five network science and three career development lesson evaluations in the summer of 2020, seven network science evaluations in the 2020–2021 academic year, and five network science and two career lesson evaluations in the summer of 2022. During the 2021–2022 academic year, many evaluations were canceled or modified using open-ended questions, given the disruptions experienced during the pandemic. Each lesson was composed of several sub-lessons/activities. Designing the lessons/activities and their objectives began by reviewing Sociedad Latina’s past curriculum materials that covered various topics of science, such as climate change, endangered animals, and deforestation, with the intention of building upon what they already had. By working with the STEAM Program Manager and STEAM Coordinator at Sociedad Latina and adding our knowledge of data science, the STEM lessons were designed based on 2–3 learning objectives that could be achieved through hands-on, interactive activities, tools, and materials. The career lessons focused on developing self-exploration and career exploration skills by helping youths become aware of who they are, what skills they possess, and how skills learned in the network science lessons transfer into the world of work. Considering that the youths were English Language Learners (ELLs), each PowerPoint slide was presented in Spanish and English. Overall, the focus of the lesson design and evaluation delivery was to make the lessons and surveys more interactive and engaging for middle schoolers by adding fun graphics and activities and reducing excessive explanations. With multiple years of experience in lesson design and evaluation, the collaborative team was able to transform the lesson and evaluation materials into fun, youth-oriented activities.

### 2.4. Measures

During the peak of the pandemic, all survey responses were collected via Qualtrics and then on paper after Sociedad Latina transitioned back to in-person teaching.

#### 2.4.1. Assessing Lesson Quality and Self-Efficacy

A quasi-experimental strategy, referred to as a separate-sample pretest–posttest design [53], provided formative assessment feedback to determine whether the network science and career lessons increased STEM learning and career development. Traditional experimental design randomly assigns individuals to receive either the intervention (i.e., STEM/career lessons) or serve as a control group by either not receiving any lessons or receiving an alternate lesson. The separate-sample pretest–posttest design enabled all youths to participate in the intervention because randomization was used to assign when—either pretest or posttest—youths were observed. Using a mixed-method strategy, this study randomly assigned youths to complete either a quantitative (self-efficacy) survey that assessed their level of confidence in engaging in the skills associated with the lesson or a qualitative question that asked more generally about their knowledge of the lesson topic. To maintain the independence of the observations, in the posttest, the youths received the alternate evaluation—those who were randomly assigned to receive the self-efficacy survey in the pretest completed the qualitative question (see Appendix B for the list of questions) in the posttest, and those who received the qualitative question in the pretest received the self-efficacy survey in the posttest.

Using a separate-sample pretest–posttest design, this study assumed the following: (a) the comparison between the pretest and posttest results assumed that any changes were due to the intervention, and (b) the separate samples assumed that the groups tested before and after each lesson were relatively similar and comparable. We acknowledge that these assumptions might limit the possibility of generalizing the results and findings (see Section 4.3).

Self-efficacy surveys in Spanish and English asked the youths to rate their level of confidence in engaging in the objectives associated with each lesson. For example, one STEM lesson objective involved the ability to describe the nature of networks. The self-efficacy item used to assess this objective was as follows: How confident are you in your ability to “Identify what types of networks exist in your life?” One career development lesson objective involved identifying one’s skills and talents. The self-efficacy item used to assess this objective was as follows: How confident are you in your ability to “Describe yourself to others.” Self-efficacy items were rated on a five-point Likert scale: 1 = “Not confident”, 2 = “Slightly confident”, 3 = “Somewhat confident”, 4 = “Fairly confident”, and 5 = “Very confident.”

A qualitative question was generated in Spanish and English for each lesson by asking the youths to reflect on their knowledge, skills, and awareness of the theme of the lesson. The qualitative question that was paired with the quantitative STEM example above was as follows: “How would you describe what a network is?” The qualitative question that was paired with the qualitative career development example above was as follows: “How would you describe yourself to others?” (see Appendix B for the list of questions).

#### 2.4.2. STEM Career Identity

In order to assess youths’ perceptions about their emerging career identities, a pretest–posttest evaluation was conducted in Spanish and English, whereby the youths were asked to respond to an open-ended survey (career narratives) at the beginning and end of the summer or after-school program. The open-ended questions included the following: (1) What are some occupational and/or life goals you are considering right now? (2) What information, knowledge, or skills do you think you need to pursue these occupational and/or life goals? (3) What are you doing right now to prepare yourself for these occupational and/or life goals? (4) What interests, skills, or values do you currently have that will help you prepare for the world of work? and (5) What are the next steps you need to take to prepare to enter one of the selected occupational and/or life goals? The career narrative items were originally developed based on the career decision phases proposed by Van Esbrock and his colleagues [54]. The specific items were selected from 18 items that were used as part of a larger national study [45].

### 2.5. Data Analysis

Two analytic methods were used in this study. To analyze the separate-sample pretest–posttest data, one-way analysis of variance (ANOVA) was used to compare the pretest and posttest responses. To analyze the career narrative data, thematic analysis was conducted to review, code, and create categories. Qualitative data were coded using NVivo software. The one-way ANOVA results provided descriptive statistics of the means, standard deviations, 95% confidence intervals, p-values, and eta-squared values. The thematic analysis provided a list of the emerging themes of career identities, consisting of goals, skills, and plans.

## 3. Results

### 3.1. Separate-Sample Pretest–Posttest Evaluation

#### 3.1.1. STEM Lesson Quality and Self-Efficacy

The impact of the program on STEM knowledge, skills, and careers was observed through the responses to both the quantitative and qualitative self-efficacy surveys. For network science, across all lessons, there were a total of 145 responses to the self-efficacy surveys collected before and after each session. One-way analysis of variance (ANOVA), comparing the pretest and posttest responses, indicated that the lessons had a meaningful, significant effect on increasing self-efficacy related to STEM learning (pretest M = 2.48, SD = 0.71; posttest M = 3.10, SD = 0.72; F (1, 144) = 27.17, *p* = 0.001, η^2^ = 0.160). An η^2^ = 0.160 can be considered a large effect size (Richardson, 2011). Table 1 provides descriptive data on the sample size, means, standard deviations, and confidence interval values.

The posttest qualitative responses demonstrated that youths were able to offer more detailed information about the lessons learned. For example, pretest responses regarding the nature of networks included the following: *“A network is like the connection of the internet”* and *“A network is a lot of people online who are connected”*. In the posttest, the responses reflected a more complex understanding: *“A network is like a large connection between things and people like maps, internet, and other things”* and *“A network consists of two or more systems that are linked in order to share resources, exchange files, or allow communications”*. For the lesson on network operating systems, the pretest responses explained only some parts of a network operating system: *“Network works when we connect 2 or more circuits”* and *“Network works by codes or programs”*. The posttest responses reflected a more comprehensive understanding of the operating process of network systems as well as a recognition of the diverse components: *“You have to have a source of solar energy, etc and that energy becomes electricity and that electricity goes through one grid to another network and that grid can be a source of energy for other things and so it goes on”*. For the lesson about the role of vaccines, the pretest responses offered explanations about the role of the vaccine but not the mechanism of how the vaccine works within the network system: *“They stop the disease from spreading”* and *“They help us with knowing if we have the virus and give us a good advice of staying at home”*. The posttest responses provided more descriptions regarding the role of vaccines and their connection to network systems: *“Vaccines help us stop the spread of a new disease because if you do give the vaccines to the person who has the most connections, you could stop the spread even more”*.

In summary, there is evidence that the network science lessons had a meaningful effect on self-efficacy. The posttest qualitative responses further demonstrated that some youths were able to provide more complex and detailed information about the nature and functioning of networks.

#### 3.1.2. MyCAP Career Lesson Quality and Self-Efficacy

For the career lessons, there were a total of 45 responses to the quantitative self-efficacy surveys. The discrepancy between a sample of 145 responses for network science and a sample of 45 responses for career lessons was due to some career lessons not being tested, given the nature of the lessons that focused on youth-led projects, civic engagement projects, guest speaker sessions, and field trips. A one-way ANOVA comparing the pretest and posttest responses found that the lessons had a meaningful, although insignificant, effect on increasing self-exploration and career exploration self-efficacy (pretest M = 3.16, SD = 0.59; posttest M = 3.35, SD = 0.37; F (1, 44) = 1.747, *p* = 0.193, η^2^ = 0.039). An η^2^ = 0.039 represents a medium effect size, which indicates that a larger sample size would likely have yielded statistically significant results [55].

Comparing the results from the pretest to those of the posttest indicated that youths focused on describing who they were at the moment of the pretest; however, youths were able to identify themselves as having multiple strengths and weaknesses, as well as how they might overcome their weaknesses after the self-exploration lesson. A pretest example includes *“I describe myself as quiet. When I warm up to others, I could speak sometimes, but I’m more of an introvert. I wouldn’t really participate as much and would probably try and disappear from existence”*. A posttest example indicates further elaborations and shows signs of motivation and ways to optimize their learning: *“I would describe myself as a person who is quiet but also willing to try new things. I love to explore the world, and I think that maybe a way for myself to come out of my shell and discover new things as well as learn from other cultures. I wanna be open but I’m also introverted as well, so it might be hard for me, but I know I’ll adapt to it quickly”*.

In summary, there is some evidence that the career development lessons had a meaningful effect on self-efficacy, whereas the posttest qualitative responses indicated that the lessons helped youths become more aware of their skills and talents and identify action steps to further develop their skills.

### 3.2. STEM Career Identity

As a summative evaluation of the overall program experience, 44 responses completed career narratives during the first and last weeks of the program (22 in the pretest and 22 in the posttest). The purpose of the career narrative analysis was to determine whether and how participation in the overall summer/academic year program that included the network science and career development lessons, as well as conversations with STEM professionals, was associated with helping youths begin to form a STEM career identity. Among the five questions, the posttest responses of youths began to reflect more complex and concrete ideas with respect to their career goals, understanding of required skills, and awareness of the current skills they possess. For career goals, the pretest responses were more general and less concrete. Some sample pretest responses include *“find my dream job”* and *“help my society develop”*. Among the 11 youths who completed the posttest in the summer of 2021, 5 provided career goal responses and 3 specifically identified STEM careers or working in STEM organizations. Based on the question about their career goals, Table 2 shows that the number of youths with the intention to pursue a STEM pathway improved after the intervention. The cohorts of 2020, 2021, and 2022 are different groups of students.

Some sample STEM-related posttest responses include “*be a pediatrician*” and *“working in NASA”*. For the required skills youths will need in order to pursue their future careers, the pretest responses were also more general, such as *“I need training”* and *“Learning a lot”*. However, more specific skills that correspond to the occupation of their choice were identified in the posttest: *“Leadership is one of the crucial [skills],”* and *“Communication skills are important for ensuring the quality message is spread throughout the organization”*. In response to the question about their current skills, youths were more likely to offer more specificity regarding their skills following the lesson. A sample pretest response is *“I’m good with kids”*, while posttest responses include *“I’m patient which will help me when studying”* and *“I know how to communicate well and more”*. In summary, the results indicate that the youths identified potential career paths and reflected more deeply on their talents and skills after program participation.

## 4. Discussion

In response to the severe under-representation of Latinx adults in STEM occupations, this community-based participatory action research project combined the efforts of a successful Latinx-serving organization with those of a university to encourage middle school-aged youths to develop STEM learning skills and consider pursuing STEM occupations (i.e., a STEM career identity). The project launch coincided with a shift to remote learning in response to COVID-19. As a summer and year-round after-school effort, the project focused on developing a strong working collaboration between Sociedad Latina staff members, who are experts in working with immigrant youths and designing culturally responsive STEM learning activities, and Boston University faculty, who provided advanced network science as well as career development lessons drawn from the Massachusetts Department of Elementary and Secondary Education’s MyCAP initiative. The aim of the study was to design and implement STEM and career development lessons that can be facilitated in a remote learning environment, as well as in person, and assess whether and how the summer and after-school learning experiences showed promise in facilitating the emergence of a STEM career identity.

This study’s findings and results are consistent with the three aspects of career development that appear in Social Cognitive Career Theory (SCCT) [14] and help us understand how they apply to the context of immigrant youths and STEM learning: (1) how STEM interests develop among Latinx middle school youths, (2) how STEM career goals in pursuing STEM careers are created, and (3) how a STEM career identity and self-efficacy that enable youths to succeed in their careers and lives are obtained. By incorporating the key ingredients of culturally responsive career development activities into the lesson design [31,32], this study also supports the importance of having Latinx role models, making connections between the skills youths possess and the world of work, and supporting youths to design personalized career plans in developing a STEM career identity. If further evaluation is conducted with a larger sample size, particularly with career lessons, it could be inferred that this study identified several important conditions associated with a STEM career identity, thereby contributing to effectively managing structural inequities in STEM fields (Manzano-Sanchez et a., 2018). This study holds promise for social transformation by confirming the findings of Blustein et al. [24] in a Latinx middle school context, with evidence that shows an increasing number of students pursuing STEM career goals and considering STEM career options.

### 4.1. Self-Efficacy and Lesson Evaluation

Deliberate efforts were made to ensure that evidence-based and culturally responsive “ingredients” for designing quality lessons were incorporated into the design [30,31,32]. In addition to translating materials into Spanish, Sociedad Latina staff created interactive elements for the PowerPoint presentations that enabled the youths to engage in discussions using video conferencing technology (e.g., Zoom, Webex).

#### 4.1.1. Formative Assessment

The network science lessons yielded a significant, large effect size, and while not statistically significant, the MyCAP lessons yielded a medium effect size related to youths’ self-efficacy, indicating a potential Type II error. While a Type I error refers to finding a chance effect, for example, conducting 100 comparisons when it is expected that 5% will be due to chance, a Type II error refers to failing to find a real effect because the sample size is too small. Therefore, given the nature of the study and the impact of COVID-19 on the low participation rates, the effect size level can be used as evidence of the promising results regarding the quality of the lessons and expectations for finding statistically significant effects for career lessons when participation rates improve. For the qualitative portion of the mixed-method design, the results also showed promising effects on the youths’ abilities to offer more complex and detailed responses to the network science subject matter, as well as more elaboration and detail about the talents and skills that are important when considering future occupations.

#### 4.1.2. Impact of COVID-19 on the Program, Process, and Youth Participation

The unprecedented circumstances caused by COVID-19 created significant challenges in conducting the project. As the reality of a pandemic set in, families scrambled to enroll their youths in our program in order to ensure they were taking advantage of opportunities for socializing and learning. As a result, initial youth enrollment was fairly successful in the summer of 2020. However, due to the technology issues families faced amid an array of varying challenges ranging from health to financial burdens, our youth retention rate was not as strong as we would have liked during the 2021–2022 academic year. Nonetheless, the youths who were able to attend and participate were fully engaged with our program. Oftentimes, the youths and families were challenged by sudden technology failures and issues that prohibited them from maximizing their participation in the program. Technology issues included poor Internet connections, computer hardware malfunctions, and burnt-out computer batteries.

Another challenge was attempting to engage with online pretest and posttest assessments within a virtual education setting. A separate-sample pretest–posttest is normally conducted “on paper” by randomly distributing a packet before students enter the classroom. Participants receive a randomly ordered Packet A or B, with Packet A including a self-efficacy questionnaire as a pretest and an open-ended question as a posttest, and Packet B including an open-ended question as a pretest and a self-efficacy questionnaire as a posttest. The physical packets minimize possible confusion among the participants and increase the overall completion rates. As the Sociedad Latina program transitioned to remote learning due to COVID-19, random assignment became one of the concerns that caused some confusion among the youths. The research team initially used a spreadsheet with three columns: Column 1 for ID numbers, Column 2 for student names, and Column 3 for the randomized assignment of the evaluation order for Types A (a self-efficacy pretest followed by a posttest qualitative item) and B (a pretest qualitative item followed by a self-efficacy posttest). The spreadsheet was stored in a separate folder due to privacy issues. The youths were verbally prompted as to which type of survey they were to complete for both the pretest and posttest. This process was deemed to be too complicated by both the STEAM coaches and the youths. For example, an examination of the survey results found that some youths selected the same type in the posttest that they completed in the pretest. As a result, modifications were made to the instructions by providing a table at the start and end of each lesson with names hyperlinked to the randomly assigned pretest and posttest so that participants could click on their names to enter the correct survey.

The online setting also created challenges for youths in being able to focus for a full 45 min on video webinar-based lessons. The initial lessons showed that youths were becoming easily distracted and showed signs of “Zoom” fatigue. The achievements of the participants were not evaluated using a format of standardized tests/scores. However, their attention and interest levels seemed to decrease based on the number of questions raised during the lessons and the limited hands-on experiences. In order to address this issue, the number of slides was reduced and more attention was given to explaining the lesson objectives and concepts that aligned with those objectives during the lesson. At the beginning of the summer of 2020, we created interactive Google slides that allowed the youths to work hands-on with the lesson content. Following professional development training on remote lesson design, we learned about Nearpod [52], a student engagement platform. Incorporating Nearpod into the online lesson delivery increased youth engagement and participation.

### 4.2. STEM Career Identity Evaluation

The overall aim of this project was to increase the number of middle school-aged youths who enter high school with the expressed intention to pursue future STEM career opportunities. The theory of change argues that youths will be more likely to consider STEM careers when they interact with STEM professionals, who serve as caring and encouraging role models, and receive advanced and relevant STEM learning and career development opportunities that enable them to become aware of how the talents and skills they possess, as well as the skills they learned during the STEM lesson, transfer to a wide range of STEM and non-STEM occupations. As a summative assessment strategy, the youths were asked to complete open-ended questions about their career goals and skills at the beginning and end of the summer or after-school program. The overall results indicate that some of the youths were able to identify more specific career objectives and were more detailed and specific about their emerging talents and skills.

#### 4.2.1. Post Hoc Evidence of an Emerging STEM Career Identity

At the end of the program, Sociedad Latina hosted a celebration for youths and their families using recorded testimonials of what they most enjoyed about the program. The network science and career lessons were a small portion of the overall programming and support offered to youths. The two youths who were asked to provide a narrative about their STEAM experiences reported the following:

“I liked to develop new networks like [we did with] the coronavirus network activity. I enjoyed learning about different types of careers, and we had some professionals from Boston University’s NEIDL [National Emerging Infectious Disease Laboratory] talking to us. I learned some skills that I may use in the future if I decide to follow a certain career. My business plan was about a company that helps old and young people with their needs of attention and more. It was fun and interesting to think about something I could be doing in the future.”

“It was fun, and I can realize a little bit about the coding world and be able to do coding on my own with Ms. Jazmin. I learned about the experiences STEM Professionals had in the past and what skills we need for that job or career. I learned a lot and saw different types of careers and jobs on O*NET with Mr. Scott from Boston University and teachers that I would study in the future like technology and other interesting things. I learned a lot of different things about jobs and careers, including Market Research Analyst and Pharmacist.”

As indicators of an emerging STEM career identity, both responses identified STEM skills as meaningful, with one youth identifying specific STEM careers of interest and the other identifying a concern they would like to address in their community. Both identified the value of engaging in STEM learning combined with interacting with professional role models. The virtual tour of the NEIDL was particularly engaging because the youths were able to learn how diseases such as COVID-19 are studied and observe the procedures and how the safety equipment is used within an actual lab setting.

#### 4.2.2. Effect of COVID-19 on Role Model Engagement

Another challenge in conducting lessons in a remote learning environment involved engaging with STEM professionals. STEM professionals in the Boston area were invited as guest speakers to talk about their career journeys and provide the youths with an opportunity to ask questions. The impact on the participants was not tested for the role model sessions. However, it was observed that their attention and interest levels seemed to have decreased, based on the number of questions asked and their overall interactions with the role models. Due to the online setting, where people often are unable to accurately understand body language and social cues, there were concerns about the quality of the conversations. In order to provide a better experience for youths, a protocol was developed to prepare the role models and the team before they engaged with the youths. Some key elements of the protocol included (a) considering role models who were more familiar with middle school students; (b) preparing a face-to-face orientation with the role models to familiarize them on how to engage with middle school youths, as well as to become familiar with the various collaboration members that facilitated the conversations; and (c) setting clear expectations and engagement parameters with youths to ensure they use the “chat” option in a professional manner.

### 4.3. Limitations

The primary limitations of this study included lower-than-expected participation rates and attempting a complex quasi-experimental evaluation strategy within a remote learning environment. With respect to participation rates, COVID-19 severely impacted access to learning for Latinx communities due in large part to the unavailability of technology, as well as the need to care for siblings, and for some, the inability to find private space for learning in their homes. Boston Public Schools and Sociedad Latina worked closely with families to ensure that laptops were available, as well as Internet access; however, there was a gap between having the technology and the ability to effectively use web-based technologies and related online learning systems.

From a research perspective, the primary limitation was a smaller-than-expected sample size that was due in large part to COVID-19. Overall, the remote learning setting also presented a number of challenges to the successful implementation of the separate-sample pretest–posttest design, which also contributed to the low number of responses received. Another challenge due to COVID-19 was the creation of remote lessons that were engaging within a remote learning environment. Such a challenge was observed through the reduced number of questions asked by the students and the interactive activities. Following the summer pilot, Sociedad Latina [56] incorporated Nearpod [52], which allows for a number of interactive features within a remote learning environment, including polling, conversation boards, and other interactive elements.

There were also limitations in relation to the assumptions underlying the separate-sample pretest–posttest design: (a) the comparison between the pretest and posttest results assumed that any changes were due to the intervention, and (b) the separate samples assumed that the groups tested before and after each lesson were relatively similar and comparable. Future research, therefore, should consider the factors that can influence the results of the pretest and posttest results (e.g., exposure to other STEM activities, the pandemic, etc.). The characteristics of each group should also be recorded and followed up to provide evidence for their comparability.

## 5. Conclusions

To address the low representation of Latinx adults in STEM professions, this community-based participatory action research study designed and piloted network science and career development lessons as part of a summer and after-school program, with the goal of increasing the number of middle school-aged youths who enter high school with the intention of pursuing a STEM occupation. The project was significantly impacted by COVID-19, which resulted in redesigning lessons, which were originally intended for in-person instruction, to be delivered using web-based technologies. Notwithstanding the loss of participation and the challenges in conducting the program assessment online, the overall results indicate the lessons showed promising results in STEM learning, self and career exploration, and the emergence of a STEM career identity, with the implication that a further evaluation is needed and can strengthen the findings with a larger sample size.

## Figures and Tables

**Table 1 behavsci-13-00949-t001:** Descriptive statistics of network science and career lesson pretests and posttests.

	N	Mean	Standard Deviation	95% Confidence Interval Lower Bound	95% Confidence Interval Upper Bound
**Network Science**					
Pretest	80	2.48	0.71	2.32	2.64
Posttest	65	3.10	0.72	2.92	3.28
Total	145	2.76	0.78	2.63	2.88
**Career Development**					
Pretest	19	3.16	0.59	2.88	3.44
Posttest	26	3.35	0.37	3.20	3.49
Total	45	3.27	0.48	3.12	3.41

**Table 2 behavsci-13-00949-t002:** Number of youths entering high school with the intention to pursue STEM careers (calculated based on the career goal responses of career narratives).

2020–2021	Non-STEM	STEM
Pretest	10	0
Posttest	4	6
**2021–2022**		
Pretest	5	0
Posttest	2	3
**2022–2023**		
Pretest	6	1
Posttest	5	2

## Data Availability

Data available on request due to restrictions eg privacy or ethical.

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
