# Peer review of "Developing STEM Career Identities among Latinx Youths: Collaborative Design, Evaluations, and Adaptations during COVID-19"

_behavsci, 2023, doi:10.3390/bs13110949_

Round 1

Reviewer 1 Report

Comments and Suggestions for Authors

This study describes a program aimed at increasing Latinx STEM Career Identity through summer and afterschool activities, including advanced STEM learning, career development activities, and access to STEM role models. The results of the program indicate positive results. The paper addresses an important problem and integrates various forms of evidence.

My main concerns revolve around the lack of clarity in presenting measures, methods, and results. Specifically:

1. The study focuses on two main outcomes: STEM learning and STEM career exploration and self-efficacy. These appear to be very different constructs. In addition, the lessons in the program included five key ingredients, such as engaging in mastery learning opportunities and having caring and encouraging adults. To enhance clarity about the underlying causal model, it would be beneficial to include a graphical representation indicating the expected relationships between components and outcomes, which can aid in understanding and replicating the study.

2. The lessons use network science as a means of fostering STEM interest. How common is network science in curricula? Are the results expected to change with other topics? Would authors recommend using network science for fostering STEM interest?

3. I would like to see, perhaps in a table, descriptive statistics regarding sample size, attrition, participants by program.

4. I would also appreciate a clearer and more detailed description of the measures used in the study, along with a definition of the construct being assessed. It was somewhat confusing to find the discussion of measures within the results section.

5. The discussion of the research method could be made clearer. The absence of a control group implies certain assumptions that should be explicitly stated. For example, the pretest-posttest comparison assumes that there are no overall time effects, meaning that any observed changes are solely due to the intervention. Additionally, the separate samples design implies that the groups receiving each assessment are comparable.  Presenting descriptive statistics of the means of each group across time would be beneficial.

Comments on the Quality of English Language

The organization of the paper could be enhanced by implementing clearer separations between sections such as theory, measures, method, program modifications due to COVID-19, and others. This would contribute to a more structured and reader-friendly presentation

Author Response

Dear Reviewer:

Thank you very much for your time and effort to review this paper. You will find our responses to each of your comments below. We have provided more detailed information on the research methods, measures, and results using descriptive statistics. Thank you, again, for your support and guidance on this project. Please do not hesitate to contact us if you have any questions or concerns.

Sincerely,

Reviewer 2 Report

Comments and Suggestions for Authors

In this paper, the authors described the development of community-based participatory activities aimed at Latino Youth in the US. The goal was to promote the intentions of the target group to pursue a career in STEM (Science, Technology, Engineering, and Mathematics). The work is interesting; however, the goals of the study are not properly defined. The results are neither convincingly presented nor sufficiently discussed. It is not clear whether career prospects have improved. The following comments should be considered:

1.The paper should be more factual. The age of the pupils should be defined. It is not enough to mention that they were „middle school-age youth”.

2.You should specify the names of Organizations 1 and 2 in the paper.

3.The paper is not properly organized. Chapters 1.2 and 1.3 describe the methodology of the research itself. As such, they should be moved to the methods section. The introduction is very short. It is only given in Section 1.1. A literature review must be extended.

4.The content of the teaching is presented on pages 5-7. It is recommended to describe the content of lessons 1-5 (network science) and summer camp activities (lessons 1-3) in tables. The inclusion of graphical elements, that is, tables and figures, will facilitate the orientation in the paper and improve the readability of the results.

5.Line numbers should be provided in the manuscript to facilitate orientation.

6.You should specify the individual items of the pre-test and post-test. They can be included in the Appendix.

7.There is a sharply decreasing number of respondents/participants when coming from the STEM lesson quality (145) through MyCAP career lesson quality (45) to STEM career identity (26). Why?

8.The impact of remote learning during the pandemic is not adequately discussed. Section 4.1.2 only describes the technical challenges, however, it does not discuss the achievements of participants. The same applies to Sections 4.2 and 4.3.

9. The aim of the study was to “increase the number of middle school-age youth entering high school” as specified on page 13 top. How did you succeed? Over the course of two years, how was the number of pupils entering high school improved?

10.The individual pupil’s responses regarding the career identity should be graphically analyzed. You should calculate the proportion of pupils who were interested in science before and after the activities. The view of science among the participants – has it been considerably improved?

Author Response

Dear Reviewer:

Thank you very much for your time and effort to review this paper. We agree with your comments and feedback and have revised our paper accordingly. You will find our responses to each of your comments below. We have provided more information on the lessons and individual items of the pre-tests and post-tests. Your comments on “the number of pupils” (Points 9 & 10), in particular, helped us reorient ourselves towards the research questions and create a clearer storyline through the manuscript. Thank you, again, for your support and guidance on this project. Please do not hesitate to contact us if you have any questions or concerns.

Sincerely,

Reviewer 3 Report

Comments and Suggestions for Authors

This article provides a clear and thorough description of a program with an important goal. However, the loss of data and degradation of methods, inflicted by the timing of program implementation coinciding with the COVID19 pandemic, are problematic for making conclusions about program outcomes. While I would hate to completely shove this one in the file drawer, I feel like I would want more data with more convincing analyses connected to them before I would reference this study in my own work. Has there been an additional cohort of students recruited over the past year whose data could be added to what's here? If so, that would certainly make the results stronger!

One specific issue: In section 3.1.2, an effect size of .039 is referred to as "large" when an eta this size is generally considered medium at best. Could this be a typo?

Author Response

Dear Reviewer:

Thank you very much for your time and effort to review this paper. You will find our responses to each of your comments below. One of the major changes we made is the inclusion of additional data focusing on career identity. Thank you, again, for your support and guidance on this project. Please do not hesitate to contact us if you have any questions or concerns.

Sincerely,

Reviewer 4 Report

Comments and Suggestions for Authors

Dear Authors,

I am pleased to review an original article draft entitled "Developing STEM Career Identity Among Latinx Youth: Collaborative Design, Evaluations, and Adaptations During COVID-19". The topic is far from being new, however, the article serves an interesting dataset with interpretation and valuable insights. 

The paper is well-written and logically structured. However, there is room for improvement:

1. You might want to structure your abstract in a more "traditional way" for better reader navigation: problem statement > aim > methods > finding > conclusions > implications

2. Also, the study objective and research questions along with explanations should be stated at the end of intro, before beginning the section 2 methodology.

3. To enhance the introduction, I would recommend:

Glebova, E. and López-Carril, S., 2023. ‘Zero Gravity’: Impact of COVID-19 Pandemic on the Professional Intentions and Career Pathway Vision of Sport Management Students. Education Sciences13(8), p.807.

4. Compelling explanations for the self-efficacy, however, it has theoretical roots in self-concept, and this point seems to be missing.

5. I would suggest to involve more pieces of RECENT literature on Social Cognitive Career Theory

6. Limitations could be extended by future research directions.

I hope you find my comments helpful.

Author Response

Dear Reviewer:

Thank you very much for your time and effort to review this paper. We agree with your comments and feedback and have revised our paper accordingly. You will find our responses to each of your comments below. We have added more recent literature on self-concept, self-efficacy, and career identity. We have also rewritten the abstract and provided clearer objectives and research questions. Thank you, again, for your support and guidance on this project. Please do not hesitate to contact us if you have any questions or concerns.

Sincerely,

Round 2

Reviewer 2 Report

Comments and Suggestions for Authors

The manuscript has been improved and is now acceptable for publication.

Reviewer 3 Report

Comments and Suggestions for Authors

Thank you for addressing the issue of lack of findings due to shortage of data! It helps to see some outcomes of the intervention.

Reviewer 4 Report

Comments and Suggestions for Authors

Dear Authors,

Thank you for the effective revisions.